# The Impact of Career Plateau on Job Burnout in the COVID-19 Pandemic: A Moderating Role of Regulatory Focus

**DOI:** 10.3390/ijerph19031087

**Published:** 2022-01-19

**Authors:** Jung Eon Kwon

**Affiliations:** Department of Career and Education Consulting, Cyber Graduate School, Joongbu University, 305, Dongheon-ro, Deogyang-gu, Goyang-si 10279, Gyeonggi-do, Korea; eoniroki@joongbu.ac.kr

**Keywords:** career plateau, job burnout, regulatory foci, COVID-19 pandemic

## Abstract

This study reviewed the mental health problems experienced by office workers exposed to new kinds of work stress, career plateau, and job burnout, due to no-contact teleworking during the COVID-19 pandemic. Human beings tend to evaluate their own qualities to determine their own superiority by comparing themselves with others. Appropriate social comparison helps to promote self-understanding and boost self-esteem. However, in the case of no-contact remote working, where the amount of time spent alone is drastically increased, the information obtained from such social comparisons is naturally insufficient, resulting in the perception of reaching a career plateau. Prolonged anxiety and a sense of helplessness have been shown to cause job burnout; however, so far, few studies have examined career plateau as an antecedent factor for job burnout. This study also considered the moderating effect of regulatory focus in order to closely examine the effect of career plateau on job burnout. According to the regulatory focus theory, differences appear in various psychological processes, such as human choices, judgments, motivations, and attitudes, determined by whether individuals adopt a promotion focus or a prevention focus. This study aimed to verify whether regulatory focus operates in a conditional context, in the process of job burnout following career plateau, to change the magnitude and direction of the influence of career plateau. To this end, a hierarchical regression analysis was performed by collecting data from 202 people working for three Korean companies. As a result of the analysis, it was found that the career plateau had a significant effect on job burnout. This direct effect was still significant even after considering the interaction with regulatory focus. In addition, promotion focus was found to have a negative moderating effect, while prevention focus had no effect on the influence of career plateau on job burnout. This study demonstrated that the negative effects of career plateau, which have been presented in various ways in academia, lead to job burnout under the non-face-to-face teleworking systems implemented due to the COVID-19 pandemic, and suggested that promotion focus can play a positive role in alleviating this dynamic.

## 1. Introduction

The economic downturn caused by the COVID-19 pandemic is leading to expanding corporate workforce restructuring. Although original restructuring was carried out with a focus on reforming and improving organizational structure, this often entailed the dismissal of personnel. However, the economic effect of layoffs on cost reduction is short-lived, and there is a growing social consensus that the negative effects on morale, cynicism, absenteeism, and turnover of the remaining organizational members after the restructuring is something to be wary of [1,2]. Therefore, by avoiding extreme restructuring methods such as layoffs, career plateau has emerged as a new mental health issue, in which organizational members find themselves stuck in a specific job position for a long time without promotion or job change.

Traditional career plateau refers to a state in which the possibility of vertical promotion or horizontal job transfer is perceived by organizational members as low [3]. In other words, the career plateau is a concept that focuses on the possibility of promotion within an organization wherein promotion is difficult or the level of promotion does not reach the socially expected level [4]. Through follow-up studies, career plateau has been developed into a concept that broadly considers restrictions on the expansion of roles and job scope within organizations, not limited to the possibility of promotion. In recent years, career plateau refers not only to situations in which the size and importance of responsibility assigned to an individual do not reflect their position, but also to situations in which the possibility of increasing challenges and responsibilities is scant due to a lack of discretion over work [5]. Therefore, managers who judge themselves as having a low contribution to the organization’s decision-making process or who feel that they have not participated in organizational innovation also experience career plateau [6].

This study focused on job burnout caused by organizational members experiencing career plateau from the perspective of social comparison theory [7]. Job burnout is a negative emotion that manifests as emotional exhaustion, depersonalization, and reduced personal accomplishment that occurs when job stressors accumulate [8]. The anxiety and sense of helplessness caused by career plateau can lead to job burnout and may be exacerbated under no-contact teleworking implemented during the COVID-19 pandemic. According to social comparison theory, human beings establish and reference standards for comparison to evaluate their abilities [9,10]. When there is an objective comparison standard, it is used, but in many cases, this may not exist, so a subjective comparison standard such as the ability of others is often applied. With the realization of no-contact telecommuting, isolated individuals may lose their comparative standards and experience more severe job burnout due to an aggravated sense of career plateau. However, there remains insufficient research regarding career plateau as an antecedent to job burnout.

This study proposed regulatory focus as a conditional variable moderating the relationship between career plateau and job burnout. This is because temperamental attributes, such as an individual’s motivation system, are highly likely to act as an individual’s interpretation mechanism in the process of perceiving career plateau and determining attitudes and actions to respond to it. Regulatory focus theory asserts that two distinct regulatory foci dominate chronic differences in individuals’ emotions and behaviors: promotion focus and prevention focus [11,12]. A person with a strong focus on promotion has a desire for growth, values hope and aspirations, and tends to focus on the benefits of success and positive outcomes rather than risks. Whereas, prevention focus refers to an inclination to cautious behavior and a strong need for safety, concentrating on avoiding situations that one does not like, and evading behaviors that deviate from common sense as much as possible [13]. Since growth and safety needs are both necessary factors for survival, promotion and prevention foci are the motivational systems of every individual.

This study considered the results of a number of previous studies showing that attitudes and behaviors differ according to the type of focus an individual has on self-regulation. It is common for people to show a proclivity towards either a prevention focus or a promotion, depending on their beliefs. The dominant focus determines the individual’s behavioral pattern in recognizing and responding to the situations they face [14]. In this study, it was expected that individuals’ coping strategies and emotional responses would be different according to their dominant type of focus, and the effects of career plateau in the process of job burnout were explored.

As mentioned above, this study explored the mental health problems experienced by organizational members in no-contact teleworking during the COVID-19 pandemic from the perspective of career plateau and job burnout. The theoretical expansion was carried out by investigating the relationship between career plateau and job burnout through the social comparison theory. In addition, this study’s attempt to examine in detail the changes in the effects of career plateau on job burnout through the concepts of regulatory focus theory is expected to substantiate suggestions for ways of overcoming job burnout and the interdisciplinary implications for academia and practice.

## 2. Theory and Hypothesis Development

### 2.1. Career Plateau

A career is the total experience that an individual develops throughout his or her life in relation to work, including attitudes and behaviors [15,16]. From an individual’s point of view, a career is an experience accumulated in the socio-economic space and time of an occupational society, and it can be said that this goes beyond just work experience, but consists of life itself [17]. Therefore, career development as a function of traditional human resource management has been treated as crucial [18].

In the early days, career development focused on the pursuit of vertical movement within an organization. This presupposes the stability of lifelong employment. However, as the employment environment has destabilized and jobs have frequently been created and destroyed, it has become difficult to maintain this premise. Accordingly, the leading subject of career planning and management is shifting from being organization-focused to being focused on the individual. In other words, the idea that organizational members should lead career development by themselves, rather than designing and managing individual career paths through individual promotion and transfer placement, is being established in the corporate field [19]. The career construction theory advocated by Savickas [20,21] can be seen as a kind of paradigm shift. He views individuals as constructing their own career paths by ascribing meaning to their career-related behaviors and professional experiences.

On the other hand, the rapid obsolescence of acquired skills and knowledge is making it difficult for the individual to maintain a stable career in one organization or one job. When the rate of career development does not keep pace with the rate of obsolescence, a career plateau may be experienced. Furthermore, due to the low-growth economy and downsizing that has become the new normal under COVID-19, members of organizations are experiencing job instability and intensified competition, increasing the prevalence of career plateau [22].

Most of the early studies on career plateau focus on the vertical and horizontal movement of jobs as a core concept. Ference, Stoner, and Warren [3] define a career plateau as the perception that the possibility of increased responsibility and authority or the possibility of promotion in the near future is slim. Veiga [23] views career plateau as a state in which the possibility of horizontal or vertical promotion is lacking. Hall [16] defines a career plateau as a state in which an individual’s current career is not commensurate with ones’ age and length of service. Near [24] uses an estimate of the time expected until the next promotion, and Chao [25] uses the time spent at the current job as an objective criterion for career plateau.

On the other hand, Ettington [26] points out that, even if an individual achieves career success such as being promoted faster than others on an objective basis, they may feel that their career has plateaued, or conversely, even if an individual has reached a career plateau according to an objective condition, he or she may not regard this as stagnant. Chao [25] advocated the concept of a “subjective plateau” and argued that the perception of career plateau can vary depending on how organizational members perceive their future careers. The noteworthy implication of Chao’s claim is that the degree of the subjectively perceived plateau can differ even if two employees are objectively in the same career plateau situation [27]. Allen, et al. [28] argued that career plateau is based on an individual’s subjective perception of future possibilities. Even the same two years may feel long to some but short to others depending on the person’s subjective perception of career plateau.

The concept of career plateau consists of a hierarchical plateau and job content plateau [29]. Hierarchical career plateau recognizes the possibility of future promotion as the essence of a plateau and appears frequently in early career plateau studies. Content career plateau is an attitude via which one feels that the possibility of increased challenge and responsibility is slim due to no longer being interested in the current job or having no discretion over the job contents [30]. This perspective approaches career plateau as a multi-dimensional construct not limited to the possibility of promotion within the organization, but also includes other job-related factors [31,32]. This means that, as suggested by Feldman and Weitz [33], a person who has increased job responsibility while still working within the same structural position for a long time may not perceive themselves as being stagnant. Yet, for the same reason, an individual promoted from a structural point of view while retaining the same level of job responsibility may view this as stagnation in terms of content.

Assuming that career plateau is observed when there are no more challenges encountered in relation to job proficiency and when a sense of pervading boredom accompanies job-related tasks, job content plateau will be at the foundation of the career plateau [19,34]. The content career plateau that employees perceive when they feel bored or frustrated because they are already proficient at their job causes them to lose curiosity and interest in their job and often feel hopeless [35]. Career plateau is linked to psychological exhaustion in which people no longer find interest in what they do [17]. Considering the characteristics of the content career plateau, it is possible that it could cause job burnout while performing no-contact teleworking due to the COVID-19 pandemic, and this is worth examining.

A perceived career plateau promotes the formation of negative attitudes such as decreased motivation for job performance, decreased job satisfaction and organizational commitment, low morale, frequent absenteeism, and elevated turnover intention [5,36,37]. The greater the difference between the ideal roles, envisioned by the members of the organization, and the reality perceived by the individual, the lower the level of job satisfaction. Conversely, the smaller the difference between the expectation of the role and the perception of the actual career, the higher the job satisfaction. The results of this study occupy the same context as a study showing a significant correlation between career growth opportunities and job satisfaction [38]. According to some studies e.g., [39], people who can no longer learn new things through a job after doing it continuously for two to five years, experience a decrease in their will to achieve.

### 2.2. Job Burnout

Job burnout generally refers to negative psychological experiences that occur as a result of repeated exposure to stress for a long time [8]. The academic concept of ‘burnout’ has been a research topic since the 1960s. The concept of job burnout related to personal work, however, was first proposed by Freudenberger [40], a psychiatrist who paid attention to a phenomenon in which medical staff, including himself, lost motivation for no obvious apparent reason. He defined it as a state in which a person experiences skepticism or frustration due to the unsatisfactory performance and reward of a given task, even though he or she has performed the task devotedly.

The study of job burnout was able to be conducted more systematically with developing MBI: Maslach Burnout Inventory [41]. Three main phenomena that negatively affect one’s mental and physical aspects, as a result of accumulating work-related stress for a considerable period of time, were accepted as a universal concept for the description and identification of job burnout. The first, emotional exhaustion, refers to the exhaustion of certain mental faculties. Mentally exhausted employees often become less adaptive and depleted of energy resulting in an inability to continue working [42]. Second, depersonalization refers to the process via which employees feel separated from the workplace or begin to show indifferent attitudes in regard to their work and duties. Job burnout refers to a state in which an individual cannot put in the energy necessary to perform his or her work-related tasks, and at the same time, to being persistently cynical with regards to colleagues [43]. Individuals experiencing job burnout tend to minimize contact with others [44]. Third, reduced personal accomplishment means that an individual no longer finds meaning in their work and that it feels meaningless to achieve what they want. Decreased sense of accomplishment as a negative evaluation of one’s competence leads to a loss of self-esteem or a decline in work ability [45].

The mental and physical exhaustion that occurs mainly in interpersonal workers in the medical or educational services industries (teachers, nurses, police, hotel workers, etc.) was studied for job burnout e.g., [46,47,48,49]. After that, the research was expanded to general occupations, demonstrating that job burnout was experienced when exposed to emotionally excessive demands for a long period of time even in occupations that deal with objects or information without necessitating the formation of close relationships between people. Job burnout is also highly related to depression and anxiety and lowers work and learning abilities [50]. In addition, job burnout increases the absenteeism rate and turnover intention, reinforcing dissatisfaction with the organization, and ultimately negatively affecting the organization [42,49].

This study hypothesized that job burnout may expand under no-contact teleworking, enacted during the COVID-19 pandemic, and the barriers to self-development and that relative deprivation caused by a perceived career plateau would affect this process. According to the social comparison theory [7], self-evaluation is not done alone, but in the context of relationships with others. That is, when individuals judge their own thoughts, attitudes, and abilities, they are constructed and evaluated through exchanges and comparisons with others [51]. From this perspective, the perception of career development and plateau is also formed by comparison with others. When telecommuting non-face-to-face and alone due to the COVID-19 pandemic, isolated individuals may lose their standard of comparison and increase their sense of career plateau. An increase in the sense of career plateau in the performance of a job strengthens the awareness that self-development can no longer be expected at work, and interaction with people in the workplace is also reduced [45,52]. Furthermore, they will experience job burnout more seriously due to the anxiety and helplessness that result from career plateau. Reflecting these arguments, Hypothesis 1 was established as follows.

**Hypothesis** **1** **(H1).***Career plateau will affect job burnout*.

There are still insufficient studies on career plateau as an antecedent context for job burnout. In order to explore these research gaps more precisely this study considered an individual’s tendency to self-regulate. Even in the same environment, the degree of job burnout will be different for each individual. This is because individual temperamental psychological factors play an important role in the recognition process of job burnout [43]. In particular, strategies for self-regulation [53] in which individuals consciously change their thoughts and behaviors in the process of pursuing goals, can affect the manifestation of job burnout as conditional moderating factors.

### 2.3. Regulatory Foci

Regulatory focus theory was advocated by Higgins [11] as a concept to explain motivational duality. According to Higgins, humans have a specific motivational system that self-regulates their actions to achieve goals. Human motivation is divided into the tendency to seek gain for pleasure and the tendency to avoid pain by preventing it in advance. This is in line with the hedonic principle, which is summarized in the two principles presupposed in traditional motivation theory: the pursuit of pleasure and the avoidance of pain [54].

This motivational system is divided into promotion focus and prevention focus [11,55]. Although both have the same goal of ultimately achieving desired end-states, the strategies, means, and methods used to achieve these end-states differ. In other words, the pattern of behavior changes according to the focus type of self-regulation, which is a specific psychological need in an individual. Promotion focus, a self-regulation type that seeks to pursue pleasure, tends to improve one’s current status in order to achieve the desired outcome and is a motivator sensitive to positive outcomes [56,57]. The focus on promotion is deeply related to advancement, growth, accomplishment, and enhancement. It is rooted in the motivation to reduce the difference between the desirable ideal self and the actual self. Promotion focus prefers eagerness, and approaching goals with a challenging attitude, despite the inherent risky [58]. Whereas, prevention focus, which is the self-regulating type that avoids pain, tends to minimize risks and losses and prevent unwanted negative consequences [59]. It is rooted in the motivation to reduce the difference between the “ought self” and the “actual self,” which values fulfillment of obligations and compliance with norms. Prevention focus is sensitive to security, safety, and responsibilities. Prevention focus aids in goal attainment by use of vigilant means to carefully identify and avoid possible problems [58].

According to the regulatory focus theory, there is a clear difference in the formation and expression of motivation in humans, and this is the result of two different psychological inclinations working together. The theory states that it is one step further from the hedonic principle of pleasure-seeking and pain-avoidance, where people self-regulate pleasure-seeking and pain-avoidance through their own strategies [12]. As a result, the regulatory focus theory encompasses the core concepts of traditional motivational theories (e.g., McClelland’s acquired-needs theory) and has a robust explanatory power for the effects of the two regulatory foci on the emotions, attitudes, and behaviors of individuals who respond accordingly [60].

The use of access and avoidance strategies can fundamentally influence information processing [61]. Therefore, people who focus predominantly on promotion focus or prevention focus show different psychological states in the process of goal achievement. That is, people with a heightened promotion focus seek after their goals eagerly, while those with a more preventive focus pursue them with vigilance. In addition, Higgins [62] reports that the intensity of regulatory focus plays a moderating role in the study of emotional response.

Self-regulation is the motivation to consciously change one’s thoughts or behavioral responses in pursuit of goals or ideals. Many previous studies e.g., [54,57,63] have demonstrated that these regulatory motivations affect a wide range of psychological processes, such as emotions, memory, and approaches to problem-solving. There are various cases where self-regulation is needed in daily life, such as resistance to temptation, suppression of impulses to give up difficult tasks, and patience while experiencing psychological pain [56]. Not expressing emotions or thoughts as they exist also requires self-regulation.

As such, self-regulation can be said to be a contextual condition that affects overall human behavior [64,65,66,67]. The effect of self-regulation, should therefore also work on changes in human behavior in the negative case of career plateau. Based on the above discussion, the research hypothesis was established that the influence of career plateau on job burnout would be different determined by the type and intensity of the regulatory foci. In particular, Hypothesis 2 regarding the promotion focus can be understood as an extension of the job characteristics model [68] that growth need play a role as a moderator in processes where the five job characteristics affect intrinsic work motivation.

**Hypothesis** **2** **(H2).***The effect of career plateau on job burnout will differ depending on the level of promotion focus*.

**Hypothesis** **3** **(H3).***The effect of career plateau on job burnout will differ depending on the level of prevention focus*.

The research model of this study demonstrating the dynamics between career plateau, job burnout, and regulatory foci are schematically shown in Figure 1. In summary, from the perspective of social comparison theory, career plateau felt by organizational members performing no-contact teleworking affects job burnout, and from the perspective of regulatory focus theory, the impact of career plateau on job burnout is expected to be different depending on the type and level of regulatory foci.

## 3. Methods

### 3.1. Participants and Procedure

To verify the above research hypotheses, a survey was conducted targeting employees of three companies that introduced remote working at the beginning of 2020, in the wake of the coronavirus outbreak. In general, a rotation method through shift organization was adopted, but these companies were implementing teleworking for all employees except for essential workers, thus meeting the criteria of this study. The industry types of companies in the study were insurance, information technology, and education service.

Data collection was conducted from 7 June to 15 June 2021. In consultation with the HR team of each company, a total of 400 questionnaires were distributed and 225 were recovered. Among them, 202 questionnaires were used for empirical analysis, excluding 23 questionnaires that were judged to have a serious central tendency or insincere responses. For empirical analysis, statistical packages SPSS 22.0 and AMOS 22.0 were used in parallel for analysis.

The general characteristics of the sample were as follows. By gender, males occupied a larger proportion of the sample; 145 males (71.8%) and 57 females (28.2%). In terms of age, 51 people were in their 20s (25.2%), 59 people in their 30s (29.2%), 56 people in their 40s (27.7%), and 36 people in their 50s (17.8%) were identified. Marital status was confirmed as 119 (58.9%) married and 83 (41.1%) single, and all respondents were identified as having received a bachelor’s degree or higher education.

### 3.2. Measures

A questionnaire instrument was used as a research tool to achieve the purpose of the study, and the items covered the major variables; career plateau, job burnout, and control focus on a 5-point Likert scale (1 = not at all, 5 = very much). Items measuring demographic characteristics were included. In order to check whether the instrument was appropriate for the employees of the research target company, a pretest was conducted for a total of 15 people, 5 from each company, from 11 January to 22 January 2021. Content validity was secured by reflecting the opinions collected in this pretest, and the items were corrected and supplemented through reverse translation by bilingual experts who were fluent in Korean and English.

#### 3.2.1. Career Plateau

To measure career plateau, the scale proposed by Milliman [69] was used. This measurement tool covers both hierarchical plateau and job content plateau. The hierarchical plateau measures the degree of promotion potential felt within the current organization, and the job content plateau measures the degree of job-related challenge rising within the current organization [3,29]. The hierarchical plateau consists of four items such as, “The likelihood that I will get ahead is limited” while the job content plateau consists of four items such as “My job tasks and activities have become routine”.

#### 3.2.2. Job Burnout

Job burnout means a psychological or emotional state in which a person becomes exhausted by the fulfillment of their job and it could drive them away from themselves from the job and others. Thus job burnout results in decreased ability to perform the job and subsequently fewer achievements. This study uses the Maslach Burnout Inventory-General Survey (MBI-GS) proposed by Maslach and Jackson [41]. MBI-GS was developed so that the Maslach Burnout Inventory (MBI), could be applied to various general workers. MBI-GS renamed the existing sub-factors such as emotional exhaustion, depersonalization, and a reduced sense of achievement into exhaustion, cynicism, and job efficacy, respectively [45,70]. A total of 16 items comprise five items of exhaustion (e.g., “I feel used up at the end of a work day”), five items of cynicism (e.g., “I doubt the significance of my work”), and six items of job efficacy (e.g., “I can effectively solve the problems that arise in my work”).

#### 3.2.3. Regulatory Foci

Promotion focus and prevention focus were measured using 18-item General Regulatory Focus Measures developed by [71]. It consists of nine items on promotion focus and the other nine items on prevention focus. The sample items for promotion focus include “I typically focus on the success I hope to achieve in the future” and sample for prevention focus are “I frequently think about how I can prevent failures in my life”.

#### 3.2.4. Control Variables

In this study on career plateau, factors judged to affect the general characteristics of respondents and job burnout were selected and set as control variables. Age, gender, and affiliated company were set as control variables. Age and tenure had a high correlation, so age was selected as a surrogate variable and the affiliated company as two dummy variables.

### 3.3. Validity and Reliability

To verify the validity and reliability of the measurement tool, confirmatory factor analysis was performed on career plateau, job burnout, and regulatory focus. Table 1 shows the results of examining the model fit index to evaluate the overall fit of confirmatory factor analysis. As a result of the analysis, it was confirmed that all major items that judge model fit, such as AGFI, TLI, CFI, and RMSEA, were at acceptable levels.

As a result of checking the factor loadings of the manifest variable for the latent variable (see Table 2), all of them except for the second item (λ = 0.493) regarding exhaustion factor was above 0.5 and significant at a 95% confidence level. It was considered appropriate convergence validity [72]. The average variance extracted (AVE) and composite reliability (CR) met the criteria (AVE > 0.5, CR > 0.7) suggested by Bagozzi and Yi [73], and Cronbach’s α was confirmed to all be 0.7 or higher, thus it could be established that there were no problems with reliability. In addition, the square root of AVE was larger than the correlation coefficients of the latent variable, thus securing discriminant validity [74].

To avoid non-response bias, the survey was conducted anonymously. The names of the metrics were not disclosed. The non-response bias was also tested by comparing the differences between the early 25% and late 25% respondents on all key variables [75,76]. There were insignificant differences in all *t*-tests (*p* > 0.10), which implies that non-response bias may not be a serious concern.

Additionally, Harman’s single factor verification [77] was conducted to confirm whether the common method bias inflated correlations. The non-rotating factor analysis was performed by the principal component method on the basis of eigenvalues greater than 1.0 for all measurement items except for control variables. As a result, seven factors were derived, and these factors were found to explain 77.26% of the total variance. The first factor, which has the most explanatory power, accounts for only 24.17% of the total variance, so it is difficult to see it as the dominant factor. Therefore, it was judged that the possibility of common method bias in this study was not a substantial threat [78].

## 4. Results

The mean, standard deviation, and correlation of latent variables are shown in Table 3. Promotion focus and prevention focus shared a negative correlation (r = −0.26, *p* < 0.01). The average career plateau was 3.73 and had a positive correlation with job burnout (r = 0.32, *p* < 0.01). This is interpreted as a result supporting, the theoretically inferred, Hypothesis 1.

A hierarchical regression analysis was performed to determine the moderating effect of the regulatory focus. Prior to this, multicollinearity occurred because the correlation between independent variables was significant. In this study’s model of variables, the variance inflation factor (VIF) of all variables ranged from 1.10 to 1.68, all below 3.0, so it was judged that there was no multicollinearity problem. The product term for verifying the moderating effect was created by converting the variable value into a deviation score [79].

The results of the regression analysis on job burnout are summarized in Table 4. Model 1 is the result of analyzing the effects of control variables. Two dummy variables were created with reference to one insurance company. Among them, a company in the education service business (Company 2) was significantly derived (b = 0.46, *p* < 0.01).

The explanatory power of model 2, which included career plateau and regulatory foci, which are the main explanatory variables of this study, was significant (R^2^ = 0.14, *p* < 0.01). The impact of career plateau on job burnout (b = 0.37, *p* < 0.01) was strong, but promotion focus (b = 0.01, *p* > 0.05) and prevention focus (b = −0.03, *p* > 0.05) was insignificant.

Model 3 is the result of adding the product term of regulatory focus and career plateau. The explanatory power was significant (R^2^ = 0.16, *p* < 0.01), and the effect of career plateau was also significant (b = 0.31, *p* < 0.05). This result supports Hypothesis 1. The effect of promotion focus, which moderates the relationship between career plateau and burnout, was significant (b = −0.38, *p* < 0.05), but the moderating effect of prevention focus was insignificant (b = 0.08, *p* > 0.05). Therefore, Hypothesis 2 was statistically supported, but Hypothesis 3 was rejected.

In order to clearly interpret the moderating effect of promotion focus, a simple slope analysis was performed as shown in Figure 2. According to the pick-a-point approach [79], a regression line was plotted at three points corresponding to ±1 standard deviation from the mean. As the level of promotion focus increases, a dampening effect can be confirmed in which the influence of career plateau on job burnout is weakened.

## 5. Discussion

### 5.1. Summary

From the perspective of reviewing the mental health problems of organizational members exposed to excessive stress under the COVID-19 pandemic, this study was pursued to identify the process that causes job burnout. The impact of career plateau on job burnout for organizational members performing no-contact telework was confirmed and the moderating effects of regulatory foci were demonstrated. To prepare a systematic rationale for this research model, social comparison theory and regulatory focus theory were applied, and data from 202 office workers were collected and analyzed from three companies that have had employees working from home for more than a year due to the outbreak of COVID-19.

As a result of analyzing the main effects of career plateau on job burnout, it was confirmed that career plateau had a significant effect on job burnout. In recent years, the COVID-19 pandemic has forced humans to accept no-contact remote working as their daily routine. According to social comparison theory, the act of evaluating one’s abilities or opinions by comparing oneself with others is a driving force to lower uncertainty and establish one’s identity. However, this is limited to non-face-to-face work from home. To make matters worse, in the aftermath of the low-growth economy, the career plateau is deepening. Under these circumstances, the negative impact of career plateau can be interpreted as exacerbating job burnout.

We presumed that the regulatory focus would moderate the relationship between career plateau and job burnout, and then verified it for each promotion focus and prevention focus. Regulatory focus is a sort of disposition formed through long experience in the process of personal growth, and is built differently for each individual. First, there was no significant correlation between the changes in job burnout according to promotion focus and prevention focus. As a result of regression analysis, it was confirmed that the direct effect of these two variables on job burnout was also insignificant. As a result of analyzing the moderating effect of these variables, promotion focus showed a significant negative moderating effect but the moderating effect of prevention focus was not significant. It can be interpreted that even job burnout caused by career plateau could be alleviated with the promotion focus. On the other hand, such a role in prevention focus could not be confirmed significantly.

### 5.2. Implications

Based on the summary of the aforementioned results, the following academic and practical implications are discussed. First, this study provides the practical implication that the negative impact of career plateau brings about job burnout under no-contact teleworking activated due to the COVID-19 pandemic. Until now, there have been few studies on the effect of a perceived career plateau on job burnout.

In addition, this study contributed to theoretical expansion by interpreting the significant correlation between career plateau and job burnout as a characteristic of no-contact teleworking, in which the instinctive desire to compare oneself with others, based on the social comparison theory, is blocked. Social comparison is used to obtain accurate information about oneself, but it tends to evaluate one’s own qualities, set personal goals, or determine whether one is superior by comparison to others. Appropriate social comparisons help promote self-understanding, boost self-esteem, and promote self-improvement. However, in no-contact teleworking, where the amount of time spent working alone increases, such social comparison information cannot be generated naturally, resulting in career plateau and job burnout due to anxiety and helplessness. Therefore, even if work is taking place on a no-contact basis, there is a need for institutional arrangements by companies to activate online meetings where social comparison is possible or to strengthen communication that provides information and feedback in a timely manner. In particular, the introduction and use of the career development system can be a way to overcome job burnout in individuals who are aware of the career plateau. It is necessary to create a practical system so that individuals can explore a career path within the organization by themselves.

Second, regulatory focus theory was applied to explore individual characteristics that can alleviate job burnout resulting from career plateau. The regulatory focus theory proposes two types of motivational mechanisms as a concept that distinguishes where an individual’s attention is focused: promotion focus that pays attention to ‘a world of gains and non-gains’ above all but preventive focus that first pays attention to and is motivated by ‘a world of non-losses and losses’. The significance of this study is that even if the level of career plateau affecting job burnout is similar, the interpretation, reaction speed, and intensity of the situation will be different depending on whether an individual is focused on promotion or prevention.

Our analysis showed that prevention focus did not play a significant role in the relationship between career plateau and job burnout, whereas promotion focus had a negative moderating effect. In other words, the higher the promotion focus, the more readily the career plateau issue was overcome, and control was implemented that prevented deterioration into job burnout. In this way, promotion focus can have a positive function in alleviating a series of negative psychological experiences and attitudes, such as various kinds of physical and mental fatigue, cynical attitude toward work, and psychological departure from work.

Promotion-focused people seek positive outcomes to achieve goals, whereas prevention-focused people are safety-oriented and seek to avoid negative outcomes in order to achieve goals. In view of this tendency, it can be inferred that, even for members who perceive career plateau, the extent to which job burnout is alleviated is stronger for the prevention-focused worker than the promotion-focused worker. Promotion focus is associated with development, growth, and achievement with the ideal goal of “this is what you ideally want to pursue”. This tendency can prevent the psychological deterioration of career plateau leading to job burnout by continuously reminding people of their career goals. Promotion-focused people are more likely to perceive more opportunities than prevention-focused people, while prevention-focused people have relatively higher risk perceptions than the promotion-focused ones, and thus take a more negative attitude toward career plateau. If a prevention-focused person concentrates on avoiding the negative consequences of a career plateau, it can be expected that they are more likely to fail in establishing progress in the direction of their career goals. In addition, if the tendency to focus on prevention is strong, a feeling of relief is felt when maintaining a successful and stable status quo, conversely, mental fatigue may occur due to a feeling of nervousness. Prevention-focused individuals are interdependent rather than independent, and thus tend to prefer getting information from others or the environment rather than themselves, which makes them more vulnerable to the negative effects of no-contact teleworking.

With these results, we confirmed the possibility that consideration of regulatory focus can be practically useful in actual organizational management. Although there is a limit to its alleviating effects on the absolute fatigue felt from the career plateau of organizational members, it was clear from our results that promotion regulatory focus serves to reduce the attitude of feeling tired of work or avoiding one’s work. According to the regulatory focus theory, an individual’s regulatory focus is established through socialization rather than being innate. In the process of socialization, different types of regulatory focus are formed depending on whether they have interacted with those who are important to them within the framework of ‘gains and non-gains’ or ‘non-losses and losses’. Therefore, situational factors emphasizing growth needs, the realization of ideals, and potential profits are highly likely to induce promotion focus. If the organizational environment or culture requires promotion focus, this means that even those with prevention focus may temporarily develop motivational psychology to pursue ideals, hopes, and aspirations. If we understand humans in terms of regulatory focus, we can effectively change an individual’s regulatory motivation through organizational culture and leadership. Such an attempt can have a positive function in alleviating the spread of negative experiences such as physical and mental fatigue and cynical attitudes toward a job that workers may feel.

### 5.3. Limitations

Along with the limitations of this study, suggestions for future research can be presented as follows. First, since this study is a cross-sectional study, it was not possible to confirm the causal relationship and the change in the relationship between variables according to the passage of time. In future research, it is necessary to investigate the phenomenon of job burnout from a longitudinal perspective by differentiating the measurement timing between the explanatory variable and the dependent variable. Second, career plateau is a concept that deals with the complex interrelationships between individuals within an organizational system, and this study is limited in that it considers only a certain set of parties of a specific company. In the future, it is necessary to enhance explanatory power by considering the influence of various stakeholders, such as superiors, subordinates, colleagues, and customers. Third, since the dummy variables of affiliated companies show a high correlation with job burnout, it is expected that the research model will be influenced by industry type. Therefore, it is necessary to expand the research subject for generalization. Fourth, we focused on work conducted at home or non-face-to-face work during the COVID-19 pandemic, but did not achieve a design to compare it with data before the COVID-19 pandemic and stayed in ex-post facto research. For follow-up studies, it is necessary to promote an experimental research design including a control group. Fifth, although common method variance was checked by Harman’s single factor test, it was not possible to directly assess the tendency for participants to respond in a socially desirable way. The self-report survey is vulnerable to social desirability bias as well as common method bias. It is likely that social desirability could be particularly salient to biased responses on the regulatory foci. We recommend future researchers use more objective methods in detecting the social desirability bias. Finally, this study was conducted based on the data of a survey conducted on three companies. It is necessary to examine whether the research results can be generalized to companies belonging to other industries. In follow-up research, we look forward to expanding to public institutions, school organizations, and manufacturing companies.

## 6. Conclusions

From the perspective of reviewing the mental health problems, this study was pursued to examine workers’ job burnout caused by career plateau under the COVID-19 pandemic. The results showed that career plateauing had a significant effect on job burnout during no-contact telecommuting work. In the process that career plateau causes job burnout, moreover, promotion focus had a negative moderating effect while prevention focus had no effect on the relationship between career plateau and job burnout. These findings, which promotion regulatory focus can alleviate a state of emotional and physical exhaustion, provide insights into where to interventions to reduce job burnout and protect the negative effects of career plateau in no face-to-face work.

## Figures and Tables

**Figure 1 ijerph-19-01087-f001:**
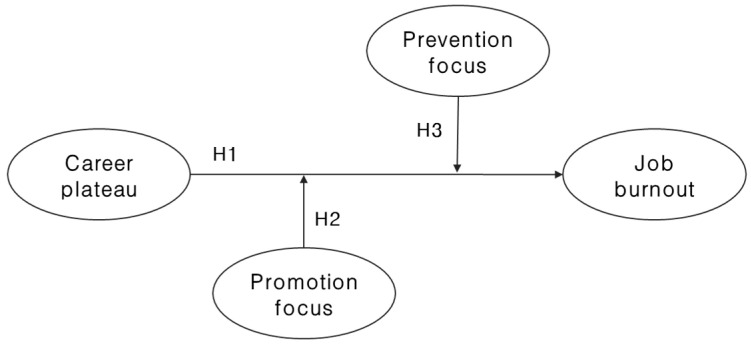
Research Model.

**Figure 2 ijerph-19-01087-f002:**
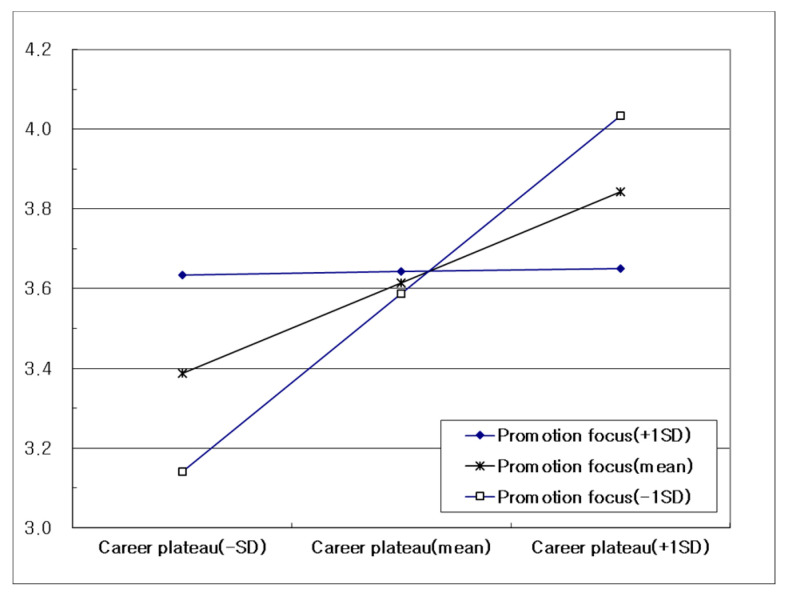
Moderating effect of promotion focus on job burnout.

**Table 1 ijerph-19-01087-t001:** Goodness of fit of the measurement model.

Index	χ^2^	df	*p*	χ^2^/df	AGFI	TLI	CFI	RMESA
Cut-off criteria	-	-	-	<2.0	>0.90	>0.90	>0.90	<0.05
Fit value	656.3	539	0.000	1.218	0.876	0.901	0.910	0.032

**Table 2 ijerph-19-01087-t002:** Reliability and validity.

Factor		Factor Loading	AVE	CR	Cronbach’s α
Career plateau	Hierarchical plateau	0.523~0.872	0.527	0.782	0.843
	Content plateau	0.588~0.917	0.686	0.820	0.927
Regulatory foci	Promotion focus	0.525~0.907	0.539	0.842	0.789
	Prevention focus	0.564~0.783	0.506	0.671	0.710
Job burnout	Exhaustion	0.493~0.897	0.534	0.693	0.751
	Cynicism	0.605~0.828	0.536	0.816	0.786
	Job efficacy	0.523~0.747	0.511	0.810	0.823

**Table 3 ijerph-19-01087-t003:** Descriptive statistics and correlation matrix.

	Mean	S.D.	1	2	3	4	5	6	7	8
1. Age	38.20	2.78	1.00							
2. Gender ^a^	0.72	0.45	0.61 *	1.00						
3. Company 1 ^b^	0.37	0.48	0.18 *	0.28 **	1.00					
4. Company 2 ^c^	0.31	0.46	0.11	0.08	0.02	1.00				
5. Career plateau	3.97	0.71	0.21 **	0.14 *	0.08	0.30 **	1.00			
6. Promotion focus	3.73	0.81	0.01	−0.03	0.06	0.19 **	0.28	1.00		
7. Prevention focus	3.59	0.93	−0.09	−0.08	−0.13	0.04	−0.35 **	−0.26 **	1.00	
8. Job burnout	3.61	1.01	0.12	0.14 *	−0.06	0.22 **	0.32 **	0.10	−0.11	1.00

Note: Sample size = 202; * *p* < 0.05, ** *p* < 0.01 (two tailed); ^a^ Dummy variables: Female = 0, Male = 1; ^b^ Dummy variables: Insurance = 0, Information technology = 1; ^c^ Dummy variables: Insurance = 0, Education service = 1.

**Table 4 ijerph-19-01087-t004:** Regression estimates for job burnout.

	Model 1			Model 2			Model 3		
	b	S.E.	t	b	S.E.	t	b	S.E.	t
Age	0.01	0.03	0.38	−0.01	0.03	−0.17	0.00	0.03	0.05
Gender ^a^	0.29	0.19	1.52	0.29	0.19	1.51	0.26	0.18	1.43
Company1 ^b^	−0.23	0.15	−1.60	−0.26	0.15	−1.81	−0.21	0.15	−1.42
Company2 ^c^	0.46	0.15	3.11 **	0.30	0.16	1.88	0.26	0.17	1.54
Career plateau (Cp)				0.37	0.11	3.40 **	0.31	0.12	2.55 *
Promotion focus (Pm)				0.01	0.10	0.06	0.03	0.09	0.32
Prevention focus (Pv)				−0.03	0.08	−0.40	−0.01	0.07	−0.16
Cp × Pm							−0.38	0.18	−2.13 *
Cp × Pv							0.08	0.21	0.37
R^2^ (adjusted R^2^)	0.08 (0.06)			0.14 (0.11)			0.16 (0.12)		
F	3.96 **			4.58 **			5.36 **		

* *p* < 0.05, ** *p* < 0.01 (two tailed); ^a^ Dummy variables: Female = 0, Male = 1; ^b^ Dummy variables: Insurance = 0, Information technology = 1; ^c^ Dummy variables: Insurance = 0, Education service = 1; Unstandardized regression coefficients reported for mean centered data.

## Data Availability

The data presented in this research are not publicly available due to participants’ privacy.

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
