# Peer review of "The Impact of Career Plateau on Job Burnout in the COVID-19 Pandemic: A Moderating Role of Regulatory Focus"

_ijerph, 2022, doi:10.3390/ijerph19031087_

Round 1

Reviewer 1 Report

Some posible minor errors:

1) Line 162, at the end: you have to remove "are".

2) Line 167, in the middle: I don't know if the verb "leaving" is here correctedly conjugated.

3) Line 353: I think you need to adjust the subject to the verb

Author Response

Some possible minor errors:

1) Line 162, at the end: you have to remove "are".

2) Line 167, in the middle: I don't know if the verb "leaving" is here correctedly conjugated.

3) Line 353: I think you need to adjust the subject to the verb

I appreciate for your careful and kind review. In addition to the errors you pointed out, we could find and modify similar errors based on your suggestions. The revised parts in this paper are marked in blue font.

Reviewer 2 Report

This study is well done,

introduction describes the problem

methods are clearly explained

analyisis is really well done

Discussion: i think this paper coud be interest for many discplines, so my only concerns is it should be more simple to read for all any stakeholders

Author Response

This study is well done,

introduction describes the problem

methods are clearly explained

analyisis is really well done

Discussion: i think this paper coud be interest for many discplines, so my only concerns is it should be more simple to read for all any stakeholders

Thank you for your kind and meaningful comments. In response to your opinion, we managed to supplement the discussion section so that stakeholders can easily understand it. The revised parts in this paper are marked in blue font.

Reviewer 3 Report

Introduction:

  1. You intruded on several studies to support the assumption regarding the the effect of career plateau on job burnout. It is interesting to research, but the serious problem is that there is no theory to support the variables in your theoretical framework. That is to say, I also can include other variables to predict job burnout in your research.
  2. The references you cited are all too old, which shows that your research issues may have been examined in many studies.

Methods:

  1. Because there is no theory to support your theoretical model, the data should be a longitudinal section to confirm the causal relationship in your theoretical model.
  2. How do you handle the no-response bias, common method bias, social-desirability bias?

Discussion

  1. You only intruded on the analysis results and proposed that the present research verified the relationship,I suggest you should rewrite the discussion to demonstrate the contributions.

Other

It is recommended that the author should reconsider the theoretical structure of your research and re-write it.

Author Response

Thank you for your careful and kind review. I could find and modify a number of shortcomings based on your suggestions. The revised parts in this paper are marked in blue font.

Introduction:

  1. You intruded on several studies to support the assumption regarding the effect of career plateau on job burnout. It is interesting to research, but the serious problem is that there is no theory to support the variables in your theoretical framework. That is to say, I also can include other variables to predict job burnout in your research.

If you could be more specific with your comment that “there is no theory”, it would help to improve this paper. We tried to investigate the relationship between career plateau and job burnout through the Social Comparison Theory and to examine the changes in the effects of career plateau on job burnout through the concepts of Regulatory Focus Theory. It was explained in Introduction sections as well as other sections. At this revision time, the regarding parts have been supplemented to make it easier to understand.

  1. The references you cited are all too old, which shows that your research issues may have been examined in many studies.

In response to your comments, some more recent papers have been added. However, even if the key variables in this paper are just only old and traditional concept, it is undoubtedly a important issues in the current corporate environment. I hope you to understand the originalities of this research that it is trying to fill a research void that has not been studied by integrating the key variables and under the special situation of COVID-19.

Methods:

  1. Because there is no theory to support your theoretical model, the data should be a longitudinal section to confirm the causal relationship in your theoretical model.

We are going to give a similar ask to you as before. If you could be more specific with your comment that “there is no theory”, it would help to improve this paper. I want you to understand that this research tried to investigate the relationship between career plateau and job burnout through the Social Comparison Theory and to examine the changes in the effects of career plateau on job burnout through the concepts of Regulatory Focus Theory.

  1. How do you handle the no-response bias, common method bias, social-desirability bias?

Using the method suggested by Armstrong & Overton, no-response bias was newly tested based on your comments and the results were described. Common method bias had been tested through Harman's single factor test and we had interpreted the bias was not serious. At this time, the explanation regarding common method bias has been supplemented to make it easier to understand. We had also thought that the unbiasedness verified in the common method test could also include that social desirability seems thus to play little role in our research model. But after receiving your advice, we realize that we had not dealt with the social desirability bias issue more detail and exactly. Thus, the consideration of social desirability bias was added as the fifth limitation of this study, explained our belated realization regarding social desirability.

Discussion

  1. You only intruded on the analysis results and proposed that the present research verified the relationship, I suggest you should rewrite the discussion to demonstrate the contributions.

As you know enough, our Discussion section consisted of 3 sub-sections: summary, implications, and limitations. Please give your specific comments regarding “discussion to demonstrate the contributions” after considering the IMPLICATIONS sub-section and it would help to improve this paper. The IMPLICATIONS sub-section including contributions has been described more clearly at this revision time.

----------------------- 

I appreciate again for your in-depth comments.